# The Impact Resistance and Mechanical Properties of Recycled Aggregate Concrete with Hooked-End and Crimped Steel Fiber

**DOI:** 10.3390/ma15197029

**Published:** 2022-10-10

**Authors:** Xiangqing Kong, Yanbin Yao, Bojian Wu, Wenjiao Zhang, Wenchang He, Ying Fu

**Affiliations:** 1School of Civil Engineering, Liaoning University of Technology, Jinzhou 121001, China; 2Songshan Lake Material Laboratory, Dongguan 523808, China

**Keywords:** steel fiber, recycled aggregate concrete, impact resistance, mechanical properties, Weibull distribution, microstructure

## Abstract

The utilization of recycled coarse aggregate (RCA) from construction and demolition waste (CDW) is a sustainable solution to protect the fragile natural environment and save the diminishing natural resources. The current study was aimed at exploring the impact resistance and mechanical properties of recycled aggregate concrete (RAC) affected by hooked-end steel fiber (HF) and crimped steel fiber (CF). Fifteen concrete mixtures considering different RCA substitution ratio, steel fiber dosage, and steel fiber shapes were designed. Meanwhile, a statistical analysis method-based Weibull distribution was introduced to evaluate the variations of impact test results, presented using a reliability function. Lastly, the microstructural morphologies of interfacial transition zones at the cement paste/aggregate and cement paste/fiber interfaces were observed using a scanning electron microscope (SEM). The experimental results showed that the impact resistance and mechanical properties mildly decreased with the increase in substitution ratio of RCA, whereas they conclusively increased with the increase in steel fiber content. Steel fiber recycled aggregate concrete (SFRAC) with 1.5% steel fiber content had the best impact resistance, and its initial cracking times and final failure times were 3.25–4.75 and 8.78–29.08 times those of plain RAC, respectively. HF has better impact resistance than CF. The SEM observations of microstructures indicated that the hardened cement paste of natural aggregate concrete (NAC) was more compact than that of RAC. Steel fiber had a better connection with the cement paste interface than that of aggregate and cement paste owing to better thermal conductivity. This research could be a guide for SFRAC as a structural material in practical engineering, steering the construction industry toward the circular economy.

## 1. Introduction

Construction and demolition waste (CDW) is solid waste produced from construction, renovation, and demotion activities [1]. Continuous industrial and urbanized development, infrastructure construction, and house building activities inevitably create huge amounts of CDW [2]. Furthermore, the increasing number of natural disasters worldwide (such as earthquakes, floods, tsunamis, and hurricanes) unavoidably cause a majority of CDW [3]. In 2018, the European Union generated 839 million tons of CDW, which accounted for 35.9% of total solid waste [4]. America produced 600 million tons of CDW [5]. China approximately generates CDW 3.5 billion tons every year, while the recovery rate is only 10% [6]. Comparatively, the recovery rate of CDW in many developed countries (USA, Singapore, Denmark, South Korea, Japan, and Germany) can reach 70–95% [7]. Globally, only 20–30% of CDW is recovered [8]. In fact, large volumes of CDW being simply filled or even illegally dumped can cause serious environment pollution including soil degradation, as well as air and water pollution. In contrast, the global construction industry is still increasing the demand for natural aggregates. According to recent statistics, global aggregate demand was 45 billion tons in 2017, but is estimated to increase to 66 billion tons in 2025 [9]. Consequently, effective and sustainable strategies for reusing and recycling CDW in the construction industry are crucial and urgent. This means that it can not only reduce environmental pollution and save land resources, but also contribute to conserving the dwindling natural resources, which is crucial for an eco-friendly society.

Previous research proposed that partial or complete replacement of natural coarse aggregate (NCA) with recycled coarse aggregate (RCA) obtained from CDW for the manufacture of recycled aggregate concrete (RAC) is a promising way to mitigate the negative impact on environment and resources. Nevertheless, RCA exhibits a high porosity ratio, with a large amount residual mortar on the surface [10]. The defects of RCA weaken the bonding between aggregate and paste by increasing the drawbacks of the interfacial transition zone. In fact, the strength of the cement matrix, the aggregates/fibers, and the interfacial transition zones between the aggregates/fibers and the matrix jointly decide the strength of concrete [11]. Accordingly, the macroscopic performance of RAC is presented as a decrease in mechanical properties [12,13,14], modulus of elasticity [15], and durability [16]. Furthermore, RAC exhibits noteworthy drying shrinkage [17,18] and creep [19]. The defects of RCA are the main engineering concern, which to some extent hinders the use of RAC in real-world applications.

It has been proven that adding fiber into RAC solves the problem of the poor mechanical properties of concrete caused by RCA [20,21]. As is well known, steel fiber is one of the most commonly used fiber materials to reduce the generation and development of concrete cracks. In the process of concrete failure, steel fiber prevents the development of cracks, whereby more energy needs to be absorbed during failure, which obviously enhances the mechanical properties, durability, and toughness of RAC. In recent years, much attention has been paid to the physical properties of steel fiber-reinforced RAC (SFRAC). Gao et al. [22] investigated the compressive behavior of SFRAC; the results indicated that the critical strain of SFRAC was improved by 29% upon adding 2% steel fiber to RAC, and the failure model changed from quasi-brittle to more ductile. The splitting tensile strength of RAC by incorporating steel fiber was explored by Ramesh et al. [23]. The experimental results demonstrated that, as steel fiber content increased from 0.3% to 0.7%, the tensile strength increased from 37.93–93%. Gao and Zhang [24] studied the flexural behavior of SFRAC and found that the flexural performance of RAC was inconspicuous for a steel fiber volume content (*V*_f_) below 0.5%, but remarkably improved when *V*_f_ increased from 0.5% to 1.0%, becoming flatter when *V*_f_ was above 1%. Afroughsabet et al. [25] studied the influence of double hooked-end steel fibers (HF) on the mechanics and durability of high-performance RAC. The experimental results indicated that adding 1% steel fiber into RAC could significantly increase the mechanical properties, while decreasing the water absorption (23%), shrinkage (15%), and electrical resistivity (86%) of the RAC. Osama et al. [26] indicated that adding HF into RAC at an amount of 2% resulted in an improvement of up to 65% in split tensile strength and a 90% improvement in flexural strength. Zong et al. [27] introduced HF into RAC to form SFRAC; experiments demonstrated that the tensile strength, peak strain, and energy absorption increased 55.90%, 48.03% and 133.1% with respect to RAC. Xie et al. [28] researched the mechanical properties of SFRAC with 1% volume content of crimped steel fiber (CF), 100% RAC substitution NCA, and different volume contents of crumb rubber substitution sand. Results illustrated that SFRAC with optimal rubber content showed good compressive and flexural behavior, highlighting SFRAC as a good compressive material in concrete engineering. Kotwal et al. [29] explored the effects of RCA and steel fiber on the properties of SFRAC at different ages; they found that 30% RCA and 0.75% steel fibers had the biggest power increase at 7, 28, and 56 days.

It should be mentioned that most of the previous experiments involving SFRAC mainly focused on its basic mechanical properties under static load, while few studies were conducted analyzing the impact resistance of SFRAC. However, the impact resistance is recognized to be an importance property of concrete in many engineering applications [30], such as airport runways, railway buffers, foundation pads of machinery, and shock absorbers [31]. Recently, Omidinasab et al. [32] found that the addition of 1% HF to RAC significantly improved the impact resistance of SFRAC and increased the ultimate energy absorption of the mixtures eightfold; moreover, HF could compensate for the weakness caused by RCA. In contrast, no further statistical analysis was performed to reveal the distribution characteristics of the impact results. Moreover, it is well known that HF and CF are two different shapes of steel fibers commonly used in engineering. However, to the best of the authors’ knowledge, investigations comparing the effect of different steel fiber shapes (i.e., HF and CF) on the mechanical properties of SFRAC, especially the impact properties under the same conditions, have not been reported. Consequently, to address the gap, the impact resistance and mechanical properties of SFRAC with HF and CF were systematically studied in this paper. Meanwhile, a statistical analysis method-based Weibull distribution was introduced to evaluate the variations and distribution characteristics of the impact test results, presented using a reliability function. Fifteen concrete mixtures considering four different steel fiber contents (i.e., 0%, 0.5%, 1.0%, and 1.5% by volume fraction), three RCA replacement ratio (0%, 50% and 100% by quality replacement rate), and two fiber shapes (HF and CF) were investigated. The compressive strength, flexural strength, and splitting tensile strength were tested and analyzed. Additionally, the microstructural morphologies of the interfacial transition zones (ITZs) at the cement paste/aggregate and cement paste/fiber interfaces were observed using a scanning electron microscope (SEM).

## 2. Experimental Program

### 2.1. Raw Materials

Ordinary Portland cement (strength 42.5 grade) was used in this study in accordance with the Chinese National Standard “Common Portland Cement” (GB/T 175-2007, 2007) [33]. Chemical and physical characteristics are shown in Table 1. River sand was used as a fine aggregate, with a fineness modulus of 2.55. Coarse aggregates (Figure 1) were NCAs and RCAs, whose physical properties are listed in Table 2. Figure 2 presents the particle size distribution curves of fine aggregates and coarse aggregates. The tested concrete specimen waste available in the laboratory was crushed using a jaw crusher (PE900 × 1200, Jinzhou, China) and screened to obtain RCAs with a particle size of 5–20 mm. Before preparing the specimens, all RCAs were soaked and then naturally dried to a saturated surface dry state. The workability of the mixtures was improved using polycarboxylate-type superplasticizer, whose water reducing ratio was 20–30%. HF and CF were used in this experiment (Figure 3). The various properties of HF and CF are summarized in Table 3. According to Zhao et al. [34], a steel fiber volume less than 0.5% has no obvious effect on the mechanical properties of concrete. However, adding more than 2% steel fiber volume content leads to a decrease in fluidity. Therefore, steel fiber content was set as 0.5%, 1.0%, and 1.5% to explore the influence of steel fiber on the mechanical properties and impact resistance of RAC.

### 2.2. Mixture Proportions and Sample Preparation

Four groups of concrete including 15 mixtures with different RCA substitution ratios (0%, 50%, and 100%), steel fiber contents (0%, 0.5%, 1.0%, and 1.5%), and fiber shapes (HF and CF) were designed, as shown in Table 4. With respect to the symbols shown in Table 4, “R”, “H”, and “C” represent RCA, HF, and CF, respectively. The numbers “0”, “50”, and “100” represent the replacement percentage of NCA by weight, and the numbers “0”, “0.5”, “1.0”, and “1.5” represent the volume fraction of steel fiber incorporated in natural aggregate concrete (NAC) and RAC. For example, R50-H0.5 denotes SFRAC with a 50% RCA replacement ratio and 0.5% HF content.

All the mixtures for testing were produced through a rotary mixer in strict accordance with Chinese regulation “Technical specification for application of fiber reinforced concrete” (JGJ/T 221-2010, 2010) [35]. The detailed mixture steps of SFRAC are as follows: firstly, coarse aggregates, fine aggregates, and cement were dry-mixed together for about 1 min. Then, steel fibers were gradually added by hand and mixed for about 1 min to ensure uniform dispersion. Lastly, the mixtures of prepared water and high-efficiency water-reducing agent were added and mixed for 2 min. After the above steps, the fresh mixtures were placed into the well-oiled mold and vibrated on the vibration table. These specimens were demolded after 24 h casting. Then, they were cured in the laboratory at 20 ± 2 °C and 95% relative humidity for 28 days.

### 2.3. Testing Methods

Test items, dimensions, and numbers of specimens used in this article are given in Table 5. On the basis of existing standards, the properties of SFRAC are discussed below.

#### 2.3.1. Compressive and Splitting Tensile Strength Tests

The compressive and splitting tensile strength was tested according to Chinese National Code “Standard for test method of mechanical properties on ordinary concrete” (GB/T 50080-2016, 2016) [36] using a universal testing machine (YAW-5000J) with measurement range of 5000 kN. Constant loading speeds of 0.7 MPa/s and 0.07 MPa/s were applied to measure the compressive strength and splitting tensile strength of the samples until failure, respectively.

#### 2.3.2. Flexural Tests

According to Chinese National Code “Standard for test method of mechanical properties on ordinary concrete” (GB/T 50080-2016, 2016) [36], at a loading speed of 0.2 mm/min, a four-point curved prismatic specimen with a span of 300 mm was subjected to load by displacement control until the specimen failed. An electronic universal testing machine (WDW-300) with a measurement range of 300 kN was record for the peak load.

#### 2.3.3. Drop-Weight Impact Resistance Tests

In conformity with the suggestion of the American Concrete Institute Committee 544 (ACI 544.2R-89, 1996) [37] and Chinese National Standard “Synthetic fibers for cement, cement mortar and concrete” (GB/T 21120-2007) [38], a homemade drop hammer impact device was used for the impact resistance test (Figure 4). A 3 kg steel ball was repeatedly pulled to 300 mm and was released with no initial speed, causing it to freefall onto the center of the specimen. The number of repeated impacts for the initial crack (*N*_1_) and ultimate failure (*N*_2_) was recorded. The percentage increase in the number of blows from initial crack to failure was labeled as the PINPB parameter [39]. The impact energy and PINPB were calculated according to Equations (1) and (2).
(1)$$W_{i}=N_{i}\cdot mgh,$$
(2)$$PINPB=\left(N_{2}-N_{1}\right)/N_{1},$$
where *W*_i_ is impact energy absorption (J), *N*_i_ is the number of blows, *m* is the drop hammer mass (kg), *g* is the acceleration due to gravity (9.8 N/kg), and *h* is the drop hammer height (mm).

#### 2.3.4. Microstructure Investigations

Through SEM (EVO-MA10, Oberkochen, Germany) observation, the microstructural morphologies of the interfacial transition zones at the cement paste/aggregate and cement paste/fiber interfaces were investigated in detail. All specimens were presoaked with anhydrous alcohol for 24 h, dried in air, and then sputtered in vacuum.

## 3. Results and Discussion

### 3.1. Mechanical Properties

#### 3.1.1. Compressive Strength

Compressive strength is an important indicator to measure the mechanical properties of concrete [13]. Figure 5 presents the compressive strength value and increment of each specimen. For the plain RAC, Rao et al. [12] and Elhakam et al. [40] reported that the reduction in compressive strength was not very prominent, when the percentage replacement of natural aggregate with RCA was up to 30% by weight. However, Behera et al. [41] reported that the decrease in compressive strength could reach 30% as compared to NAC at 100% replacement with RCA. As expected, the compressive strength of the concrete gradually decreased with increasing RCA replacement ratio. The compression strength of plain RAC with 50% and 100% RCA replacement rate decreased by 4.0% and 7.8% compared to NAC, respectively. This indicates that RCA had an adverse effect on the compressive strength, which is similar to previous findings [13,42]. This phenomenon may have been caused by the lower apparent density and higher crushing index, whereby the crushing index affects the compressive strength. A large crushing index is usually associated with a lower compressive strength [27]. The water absorption [20] of RCA also weakens the bearing capacity of the RAC matrix. Actually, during the preparation of RCA, more defects such as holes and initial cracks are produced inside the aggregate, which often lead to stress concentration at the cracks tip under load, thus affecting the strength of concrete. In addition, as can be seen from Figure 5, all SFRAC specimens exhibited higher compressive strength than the control specimen (RAC without steel fiber), and the compressive strengths gradually increased with increasing steel fiber content. In particular, when adding 1.5% steel fibers into RAC, mixture R0-H1.5 achieved the maximum compressive strength of 56.2 MPa. The compressive strength decreased slightly with RCA substitution of NCA from 0% to 100%, whereby mixtures R50-H1.5, R100-H1.5, and R100-C1.5 attained 54.3 MPa, 53.2 MPa, and 51.9 MPa, respectively, which were 19.9%, 22.3%, and 19.3% higher than common RAC. There are two reasons for the improvement of compressive strength: (1) the high Young’s modulus and stiffness of steel fiber contributed to an increase in the compressive strength of the concrete matrix; (2) as shown in Figure 6, the random distribution of fibers in concrete provided a confinement effect, which restricted the crack propagation [43,44]. Furthermore, comparing the SFRAC specimens with a 100% RCA substitution rate, it can be observed that HF had a better influence than CF on the compressive strength of RAC. This can be explained by the outstanding anchorage force and the friction between HF and the concrete matrix, which produced a boon effect in contrast to CF, thereby serving to provide SFRAC with lateral confinement under compression. Moreover, the geometric size of HF was smaller than that of CF; hence, the fiber spacing of HF was lower at the same volume fraction and substantially improved its dispersion, resulting in more defects in the concrete matrix. Yao et al. [45] pointed out that the matrix can be strengthened by reducing the specific spacing of fibers.

#### 3.1.2. Splitting Tensile Strength

The splitting tensile strength is a significant factor for measuring the cracking resistance of concrete, which tremendously influences the durability and safety of the concrete structure. Figure 7 shows the splitting tensile strength value and increment of each specimen. As can be observed, the splitting tensile strengths gradually decreased with increasing RCA replacement rate. The splitting tensile strength of the plain RAC specimens with a 50% and 100% RCA replacement rate decreased by 9.5% and 15.4% compared to NAC, respectively. The decreases were characterized by an increase in both porosity and interfacial weak areas as a result of the increase in RCA ratio, due to the same factors mediating compressive strength, as described above. Upon adding 0.5–1.5% steel fibers into RAC, the splitting tensile strength increased by about 10.5–34.6%, respectively. The maximum splitting tensile strength was 4.24 MPa upon adding 1.5% steel fiber into NAC. The splitting tensile strength of mixtures R50-H1.5, R100-H1.5, and R100-C1.5 increased by 33.3%, 34.6%, and 32.2%, reaching 4.08 MPa, 3.85 MPa and 3.78 MPa, respectively. Obviously, the presence of steel fiber dramatically improved the splitting tensile strength of RAC. It is well known that the strength of concrete mainly depends on the strength of the cement matrix and aggregates, as well as the interfacial transition zone (ITZ) between the aggregates and the matrix [46]. However, for fiber-reinforced concrete, the fiber component is used to retard the appearance of macro-cracks before reaching the critical flaw size, which in turn increases the splitting tensile strength. Additionally, comparing the SFRAC specimens with a 100% RCA substitution rate, it can be found that HF had a better effect than CF on the splitting tensile strength of RAC. This was mainly attributed to HF having a higher bridging capacity and crack resistance than CF, as shown in Figure 6.

#### 3.1.3. Flexural Strength

Flexural strength, also known as the rupture modulus, is another widely used measure of tensile strength [43]. Figure 8 plots the flexural strength test results of all SFRAC specimens as a function of fiber content. It shows that the flexural strength with a 50% and 100% RCA replacement rate decreased by about 3.5% and 6.0% compared to NAC, respectively, indicating that the flexural strength slightly decreased with increasing RCA substitution ratio. Mohammed and Najim [14] also pointed out that the flexural toughness of self-compacting RAC beams was lower than that of NAC beams, and the ultimate deflections of beams increased with increasing RCA replacement ratio. This may be due to the strength of RCA being weaker than that of NCA, while the bonding strength of the RCA surface and new mortar is weaker than that of NAC [47], resulting in accelerated crack propagation. The incorporation of steel fiber could considerably enhance the flexural strength of RAC, with an improvement rate of 5.8–35%. The maximum increase in flexural strength at different RCA replacement rates was also achieved with a steel fiber content of 1.5%. The flexural strength of specimens R0-H1.5, R50-H1.5, R100-H1.5, and R100-C1.5 increased by 32%, 34%, 35%, and 28%, reaching 7.50, 7.34, 7.21, and 6.83 MPa, respectively. In fact, as mentioned above, due to the bridge function of steel fiber, the micro-cracks in concrete could fully play the role of a bridge and enhance the flexural strength of RAC. Additionally, comparing the test results with 100% RCA substitution rate and different steel fiber shape, it can also be found that the effect of HF was more obvious than that of CF on flexural strength. Actually, the mechanical performance of fiber-reinforced concretes mainly depends on the fiber length, aspect ratio, and tensile strength [48]. The finer and thinner size of HF can be more easily dispersed into concrete and form the three-dimensional network structure, potentially resulting in a stronger bond between HF and concrete than between CF and concrete. A corresponding explanation for this phenomenon can also be found below in the investigation into microstructures of the ITZ between cement and fiber.

### 3.2. Impact Resistance

#### 3.2.1. Impact Test Results

Impact resistance is the ability of materials to absorb kinetic energy under repeated loads [39]. The improved impact resistance is one of the important aspects of fiber-reinforced concrete [49]. Table 6 and Table 7 indicate the influence of different RCA ratios, steel fiber contents, and shapes on the impact resistance of RAC. The results indicate that the impact resistance capacity of SFRAC gradually decreased with increasing substitution ratio of RCA. The initial crack impact energy of plain RAC specimens R100-H0 and R50-H0 decreased by 25% and 33% compared to specimen N0-H0, while the failure impact energy also decreased by 15% and 31%, respectively. Additionally, the blow numbers of the initial crack of plain RAC were nearly the same as those of ultimate failure, demonstrating the high brittleness of the plain RAC. As can be seen in Figure 9, the incorporation of steel fiber could improve the toughness of RAC, while the impact resistance improved significantly with the increased content of steel fibers. Specifically, RAC specimens R0-H1.5, R50-H1.5, R100-H1.5, and R100-C1.5 containing 1.5% steel fibers achieved the highest initial crack impact resistance, which was 4.75, 5.67, 6.13, and 3.25 times higher than the plain RAC specimens, respectively. Moreover, the failure impact resistance of RAC specimens R0-H1.5, R50-H1.5, R100-H1.5, and R100-C1.5 increased by about 29.08, 27, 22.89 and 8.78 times with respect to plain RAC, respectively. Figure 9 illustrates the influence of steel fiber content on the PINPB. As can be observed, the PINPB decreased with the increasing substitution ratio of RCA, but was significantly improved upon adding steel fiber. The reason may be that steel fiber provides a micro-reinforcement system for the RAC matrix, which produces a spring-like buffering effect under impact load and effectively delays the formation and development of cracks; accordingly, the failure mode of concrete changes from brittleness to elastic–plastic failure. Compared with the SFRAC specimens with a 100% RCA substitution rate, it can be observed that HF had a better effect than CF on impact resistance in terms of the blow numbers for initial crack (*N*_1_) and ultimate failure (*N*_2_). The reason may be that HF provided better fiber anchorage than the CF, helping the fibers to transfer a higher level of stress and improve their capacity in delaying the development of cracks [50]. The explanations given above can be proven from the impact failure pattern of RAC and SFRAC specimens, as presented in Figure 10a–f. Figure 10a,b indicate that plain RAC specimens suffered the most serious damage under impact load with poor integrity, presenting typical brittle characteristics. The cracks expanded rapidly upon appearing, and the specimen lost its bearing capacity quickly, finally breaking into several parts. However, as shown in Figure 10c–f, the failure morphology was significantly improved after steel fiber was incorporated, whereby the development of cracks changed from a few larger geometric shapes to longer and thinner patterns, and the specimens maintained good integrity after the failure. In addition, the SFRAC specimen could still bear the impact load after cracks appeared. The cracks developed slowly from the center to the edge of the specimen, and they penetrated from the bottom of the specimen to the upper surface. By comparing Figure 10c,d and Figure 10e,f, it can be found that the failure crack length and width of the HF specimens were longer and thinner than those of the CF specimens, while the crack resistance effect is better. This is in line with the experimental data, confirming that HF has a better impact resistance than CF. It should be noted that there was some dispersion in the impact experimental data, which may have been caused by the following reasons: (1) due to manual operation, the height of the fall could not be precisely controlled; (2) the impact strength results were obtained from a single impact point on the concrete surface, which may have been the solid surface of a coarse aggregate or the soft layer of concrete or fiber; (3) concrete is a heterogeneous material, which can directly cause a change in the design mix ratio, resulting in a change in impact strength [51,52]. In view of the features of the impact test results, statistical analysis emerged as an appropriate solution for resolving the variations in the impact test results.

#### 3.2.2. Weibull Distribution

Diverse probability models have been presented in the past few decades and applied for statistically analyzing the experimental results of concrete experiencing low-speed repeated impact tests. Among them, the efficiency and accuracy of the two-parameter Weibull distribution for evaluating the impact performance of concrete has been proven by some investigations [30,53,54,55,56]. Therefore, this paper adopted a two-parameter Weibull distribution to conduct a statistical interpretation of the SFRAC impact test data.

Equation (3) shows the probability density distribution function *f*(*n*) of the two-parameter Weibull distribution.
(3)$$f(n)=\frac{\alpha }{u}{\left(\frac{n}{u}\right)}^{\alpha -1}\exp\left[-{\left(\frac{n}{u}\right)}^{\alpha }\right].$$

According to Equation (3), the cumulative distribution function *F*(*n*) can be obtained using Equation (4).
(4)$$F(n)=1-\exp\left[-{\left(\frac{n}{u}\right)}^{\alpha }\right],$$
where *n* is the concrete value of impact life (i.e., *N*_1_ and *N*_2_ in the current paper), *α* is the shape parameter (i.e., Weibull slope), and *u* is the scale parameter. Saghafi et al. [57] proposed that the probability of survivorship function can be expressed as
(5)$$L(n)=1-F(n)=\exp\left[-{\left(\frac{n}{u}\right)}^{\alpha }\right].$$

Taking the natural logarithm against both sides of Equation (5) twice yields Equation (6).
(6)$$\ln\left[\ln\left(\frac{1}{L(n)}\right)\right]=\alpha \ln(n)-\alpha \ln(u).$$

Setting *Y* = ln[ln(1/(*L*(*n*))], *X* = ln(*n*), and *β* = *α*ln(*u*), Equation (7) can be obtained.
(7)Y=αX−β.

Thus, Equation (7) can be used to verify the linear correlation of *Y* and *X*. The impact property data *N*_1_ and *N*_2_ are listed in ascending order. According to Ding et al. [30], the survivorship probability function *L* can be represented by Equation (8).
(8)$$L=1-\frac{i}{k+1},$$
where *i* represents the order number of blows in sequence, and *k* denotes the total number of impact samples in each group.

Next, it is necessary to verify whether there is a linear correlation between *Y* and *X*. If there is a linear correlation between X and Y, the two-parameter Weibull distribution is suitable to describe the impact resistance. Furthermore, the regression coefficients *α* and *β* and correlation coefficient *R*^2^ were obtained through regression analysis. The data distribution and fitting curve of the test data for the number of impacts on *N*_1_ and *N*_2_ are shown in Figure 11 and Figure 12, respectively. The regression coefficients *α* and *β* and correlation coefficient *R*^2^ for *N*_1_ and *N*_2_ are given in Table 8. There was an obvious linear correlation between *Y* and *X*. Figure 11 and Figure 12 demonstrate that the two-parameter Weibull distribution was appropriate to describe the impact resistance of *N*_1_ and *N*_2_. Moreover, as can be seen in Table 8, the minimum value of *R*^2^ was 0.806, while most values were above 0.9, revealing adequate reliability. Rahmani et al. [58] and Ali et al. [54] also pointed out that a correlation coefficient greater than or equal to 0.7 is adequate for a two-parameter Weibull distribution. Here, the correlation coefficient *R*^2^ values were greater than or equal to 0.8, validating the two-parameter Weibull distribution as a proper approach to depict the statistical distribution of the impact test data for SFRAC specimens.

Rearranging Equation (6) and substituting (1 − *P*_r_) for *L*(*n*), the number of blows to *N*_1_ and *N*_2_ of all mixtures under different failure probabilities can be obtained as follows:(9)$$n=\ln^{-1}\left\{\frac{1n\left[\ln\left(1/(1-P_{r})\right)\right]+\alpha \ln(u)}{\alpha }\right\},$$
where *P*_r_ is the failure probability.

According to Equation (9) and the regression parameters in Table 8, the calculated values of impact life *N*_1_ and *N*_2_ of SFRAC specimens under different failure probabilities (*P*_r_) were obtained, as shown in Table 9. The findings indicate that the impact life of SFRAC specimens increased with the increase in failure probability, conforming to the physical characteristics of concrete materials. The impact lives of plain RAC were low and not notable under different failure probabilities. Under the same failure probabilities, the estimated *N*_1_ and *N*_2_ of each group of specimens were consistent with the trend of the impact test results. Specifically, with the increase in RCA replacement rate, the impact life gradually decreased. At the same RCA replacement rate, the impact life increased with the increase in steel fiber content. By comparing the statistical analysis results in Table 9 with the impact test results in Table 6 and Table 7, it can be found that the statistical analysis results were close to those of the impact tests, converging as the failure probabilities increased, thus proving the validity of the Weibull distribution. According to the results, the *P*_r_–ln*N*_1_–*V*_f_ and *P*_r_–ln*N*_2_–*V*_f_ relationship curves could be obtained for each mixture under different failure probabilities, as shown in Figure 13 and Figure 14, respectively. From these reliability curves, the impact resistance of SFRAC could be predicted accurately according to the steel fiber content with consideration of the failure probability. This statistical tool can eliminate the additional cost and time to assess the impact strength of SFRAC specimens using the repeated drop hammer impact test. For engineers and designers, on the other hand, this statistical tool can help in the selection of a reliable impact strength [59,60]. Furthermore, it can help reinforced RAC material producers confidently present the required mechanical properties to the consumer.

### 3.3. Microstructure of Steel Fiber-Reinforced RAC

It is known that concrete is a typical heterogeneous material, and its strength mainly hinges on the cement-based strength and the bonding characteristics of the ITZ between the aggregate/fiber and the matrix [11,61]. Although the ITZ only accounts for a small proportion of hardened concrete, it has a remarkable effect on the physical and mechanical properties of concrete [39]. Therefore, scanning electron microscopy (SEM) was applied to seven group samples, including plain RAC specimens with RCA substitution rates of 0%, 50%, and 100%, as well as SFRAC specimens reinforced with HF and CF. The microstructure of hardened cement slurry and the ITZ between aggregate/fiber and cement slurry were investigated.

#### 3.3.1. Microstructure of Hardened Cement Paste

Figure 15a–c present the microstructure images of hardened cement pastes with different RCA substitution rates of 0% (R0-H0), 50% (R50-H0), and 100% (R100-H0), respectively. As can be seen in Figure 15, the microstructure of cement slurry in specimen N0-H0 was the most compact and homogeneous, and, except for a small number of unhydrated particles, it was difficult to observe the existence of harmful micro-cracks and pore zones. Owing to the good hydration response of cementitious materials, these unhydrated cement particles were tightly wrapped by integrated calcium silicate hydrate (C–S–H) gels, forming a solidified gel bonding system (Figure 15a). From Figure 15b, it can be seen that the surfaces of hardened cement paste in specimen R50-H0 looked much rougher, and there was more clinker dispersed in the internal structure as compared to specimen N0-H0. Meanwhile, many harmful gels (not embedded with the calcium hydroxide (Ca(OH)_2_), whereby C–S–H gels formed a continuous phase) were noticeable within this hydrated paste, causing the hardened cement slurry to become loose. The reason for this phenomenon may be attributed to the pre-wetting before RAC specimen preparation, which led to the actual water cement ratio being larger than the theoretical value, thereby affecting the continuity of hydration products. The structural looseness caused by the high water–cement ratio was one of the reasons for the decrease in RAC strength. As can be seen in Figure 15c, there were a large number of unhydrated clinker particles and dispersed cementitious particles on the hydrated cement paste surface of the R100-H0 specimen. In addition, obvious holes and wide cracks existing in the inner structure were found. The same trend was also evidenced by Dong et al. [62], whereby the density of the RAC slurry decreased but the pores increased with increase in RCA substitution ratio. However, the strength of concrete mainly depends on the strength of cement matrix and aggregate. The increase in holes and cracks led to a decrease in RAC density and structural looseness, which negatively affected the strength of the cement paste.

#### 3.3.2. Microstructure of the Aggregate–Paste Interface

Figure 16 shows the microstructure of the ITZ between the cement slurry and aggregates. For the NCA specimen, the hydrated cement paste adhered to the NCA, featuring many well-developed hydration products and gel components. Furthermore, these gels looked more complete and more continuous, and they had a higher strength and density, which rendered the interface between aggregate and cement paste relatively fuzzy. Because of the nonuniform volume deformation and shrinkage of the cement paste caused by the change in water content during the hardening process, inner stress was induced in the interfacial area leading to the separation of the cement paste from the aggregate, as shown in Figure 16a. The ITZ between the cement paste and RCA due to the adhesion of the old mortar to the RCA was not as dense as the NAC, while it was more obvious than that of the NCA specimen, as shown in Figure 16b. There were a large number of pores and dispersed gel particles observed in the interfacial zone of the RAC specimen, which seriously weakened the bonding performance between the aggregate and cement paste. This indicates that the ITZ between them was weaker. It is well known that the ITZ between the aggregate and matrix is an important factor influencing the strength of concrete. The loose and porous ITZ structure between RCA and cement paste led to a relative decrease in the macro-mechanical properties of RAC, i.e., the compressive strength, flexural strength, and splitting tensile strength.

#### 3.3.3. Microstructure of the Fiber–Paste Interface

Figure 17 shows the microstructure images of the ITZ between cement slurry and fibers. It can be observed that the steel fiber had a good connection with the cement paste interface, and that there was no obvious transition zone around the fiber, which presented better bonding properties as compared to the microstructure of the ITZ between cement slurry and aggregates (see Figure 16). It can be clearly observed from these pictures that the cement composites around the steel fiber contained many gel components, and that there was adequate hydration. Fully hydrated cement composites become more compact, thus improving their strength. It was also seen that some of the cement pastes on the side surface of the steel fiber were embedded into the surface. This demonstrates that a chemical reaction between cement pastes and steel fibers occurred, rendering the interfacial zone dense and integral. The better bonding performance and the dense ITZ between the fiber and the cement paste would result in fiber pullout requiring more energy, which was one of the reasons for the improvement in mechanical properties and impact resistance. The significant improvement in interfacial microstructure can be attributed to the heat diffusion influence of steel fibers, which serves to promote the hydration reaction of surrounding pastes [61]. Furthermore, it can be observed from Figure 17a that the surface of HF was covered with dense hardened cement paste. Villiform crystals and other discontinuous gels can hardly be observed. However, as can be seen from Figure 17b, the hardened cement paste covering the CF surface contained multiple crystals, and a small pore zone can be noted among these crystalline pastes, which did not form a complete and continuous layered structure. This may be because the wavy surface of SF increased the water retention capacity of the fiber surface in the hydrated cement matrix, resulting in a worse bonding performance of CF compared to HF. This indicates that less energy would be absorbed in the event of damage, explaining why the HF-enhanced RAC had better mechanical properties and impact resistance compared to CF-enhanced RAC.

## 4. Conclusions

The impact resistance of SFRAC with different RAC replacement rates and different steel fiber contents was tested and described, and the results were analyzed using a Weibull distribution. In addition, the microstructures of SFRAC were investigated using SEM. According to the experimental results and analysis, detailed conclusions can be drawn as follows:

The mechanical properties of RAC gradually decreased with increasing RCA substitution ratio. When the substitution ratio of RCA increased from 0% to 50% and 100%, the compressive strength, flexural strength, and splitting tensile strength of RAC decreased by 4–7.8%, 9.5–15.4%, and 3.5–6%, respectively.The incorporation of steel fiber significantly increased the mechanical properties of RAC, which could be attributed to the high Young’s modulus, stiffness, and bridging capacity of steel fiber. Specifically, upon adding 1.5% steel fiber into the RAC, mixtures R0-H1.5, R50-H1.5, R100-H1.5, and R100-C1.5 achieved the maximum compressive strength, flexural strength, and splitting tensile strength, which were 19.1–22.3%, 32.2–34.6%, and 28–35% greater compared to common RAC, respectively.The impact resistance of SFRAC decreased with increasing RCA substitution ratio, but significantly increased with steel fiber content. Among the different steel fiber combinations, the SFRAC sample containing 1.5% steel fiber had the maximum impact resistance, and its numbers of blows (*N*_1_ and *N*_2_) were 3.25–4.75 times and 8.78–29.08 times greater compared to the plain RAC specimens, respectively. The improvement effect of HF on the impact resistance of RAC was more significant than that of CF.The correlation coefficient *R*^2^ values obtained following the statistical analysis of the impact performance of concrete were above or equal to 0.806, exceeding the recommended limit value reported by previous research, which indicated that the distribution characteristics of SFRAC impact obeyed the two-parameter Weibull distribution.With increasing RCA replacement ratio, the density of the hardened cement paste decreased but the pores increased. There were a large number of pores and dispersed gel particles observed in the ITZ between cement paste and RCA, and the ITZ between the RCA and cement paste was more obvious compared to the NCA specimen. The connection between steel fibers and cement paste was better than that between the aggregates and cement paste. The wavy surface of CF increased the water retention capacity of the fiber surface in the hydrated cement matrix, resulting in a worse bonding performance of CF compared to HF.

## Figures and Tables

**Figure 1 materials-15-07029-f001:**
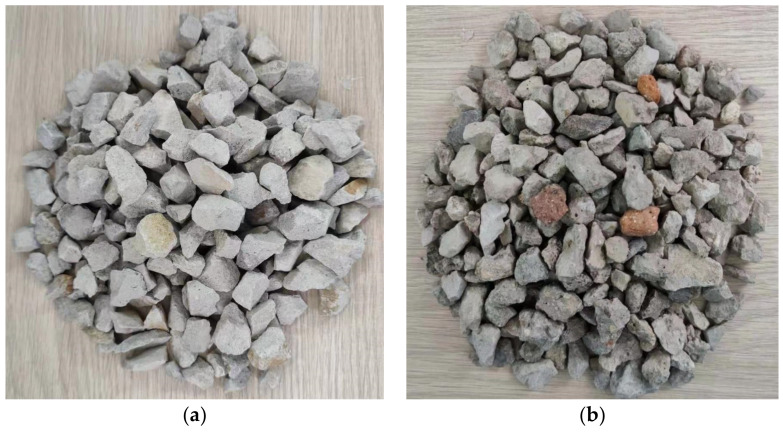
(**a**) Natural coarse aggregate; (**b**) Recycled coarse aggregate.

**Figure 2 materials-15-07029-f002:**
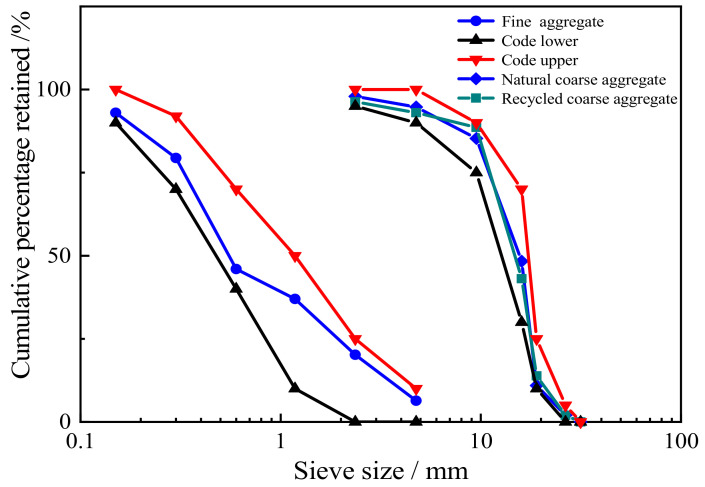
Gradation curves of aggregates.

**Figure 3 materials-15-07029-f003:**
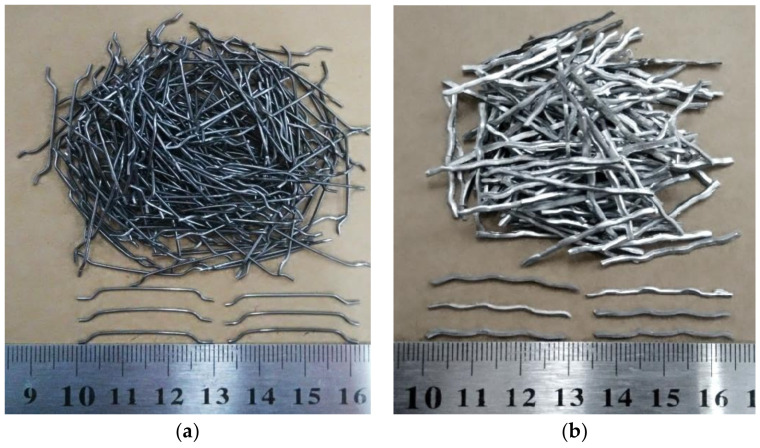
(**a**) Hooked-end steel fiber; (**b**) Crimped steel fiber.

**Figure 4 materials-15-07029-f004:**
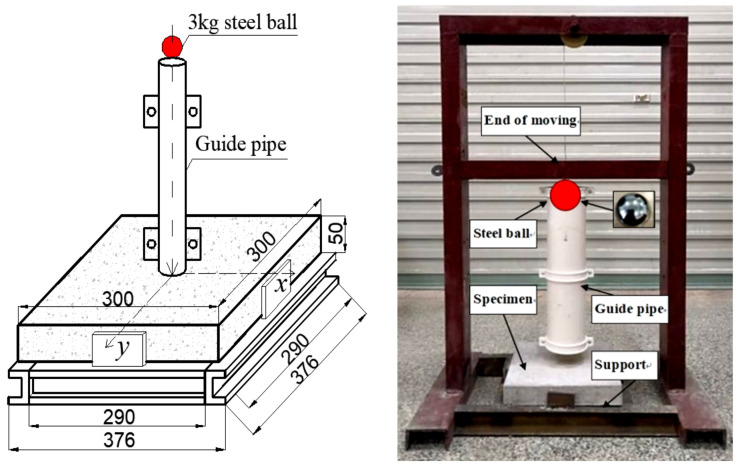
The drop weight impact test device of concrete.

**Figure 5 materials-15-07029-f005:**
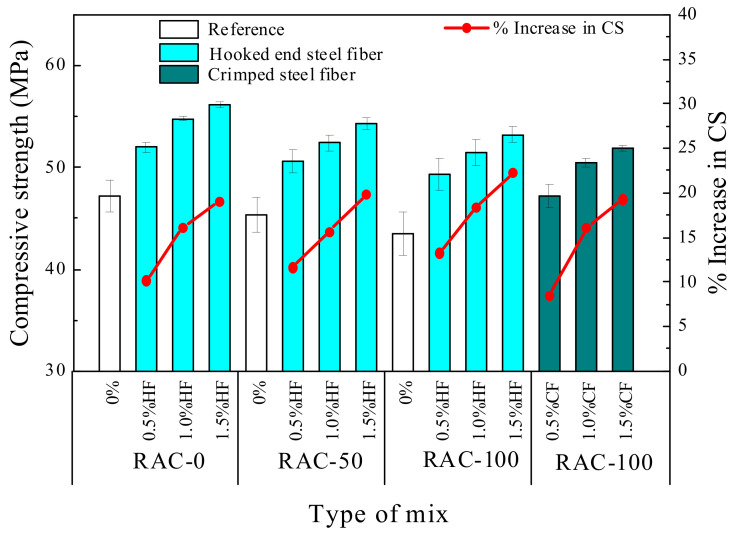
Compressive strength of fiber-reinforced RAC specimens.

**Figure 6 materials-15-07029-f006:**
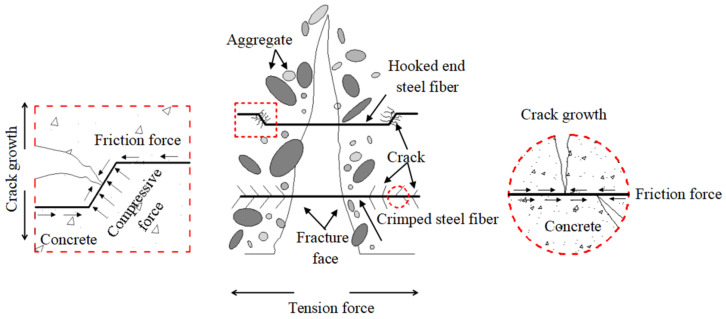
Crack resistance and strengthening model of steel fiber-reinforced concrete.

**Figure 7 materials-15-07029-f007:**
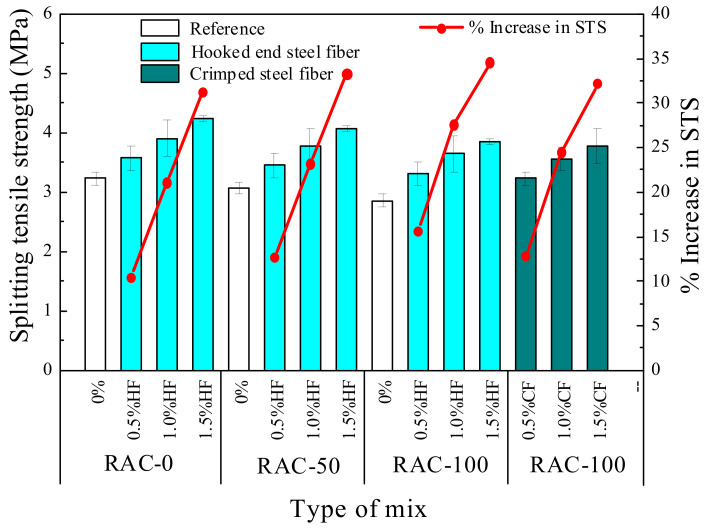
Splitting tensile strength of fiber-reinforced RAC specimens.

**Figure 8 materials-15-07029-f008:**
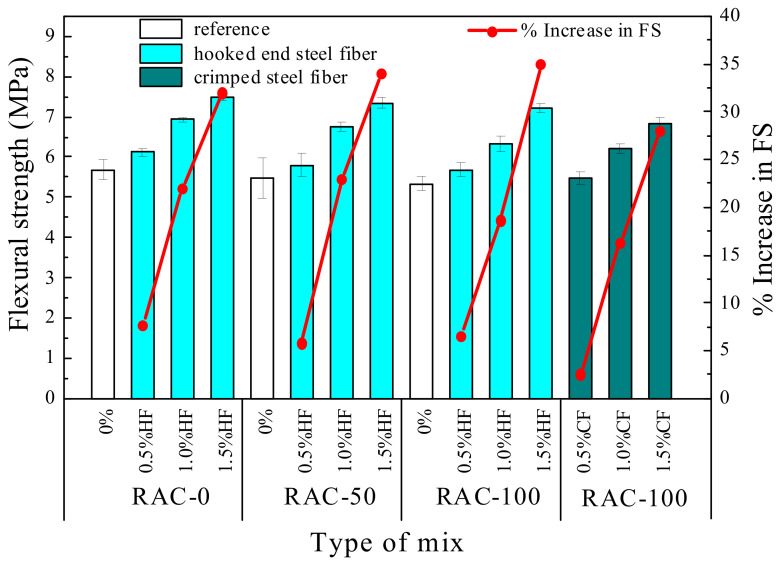
Flexural strength of fiber-reinforced RAC specimens.

**Figure 9 materials-15-07029-f009:**
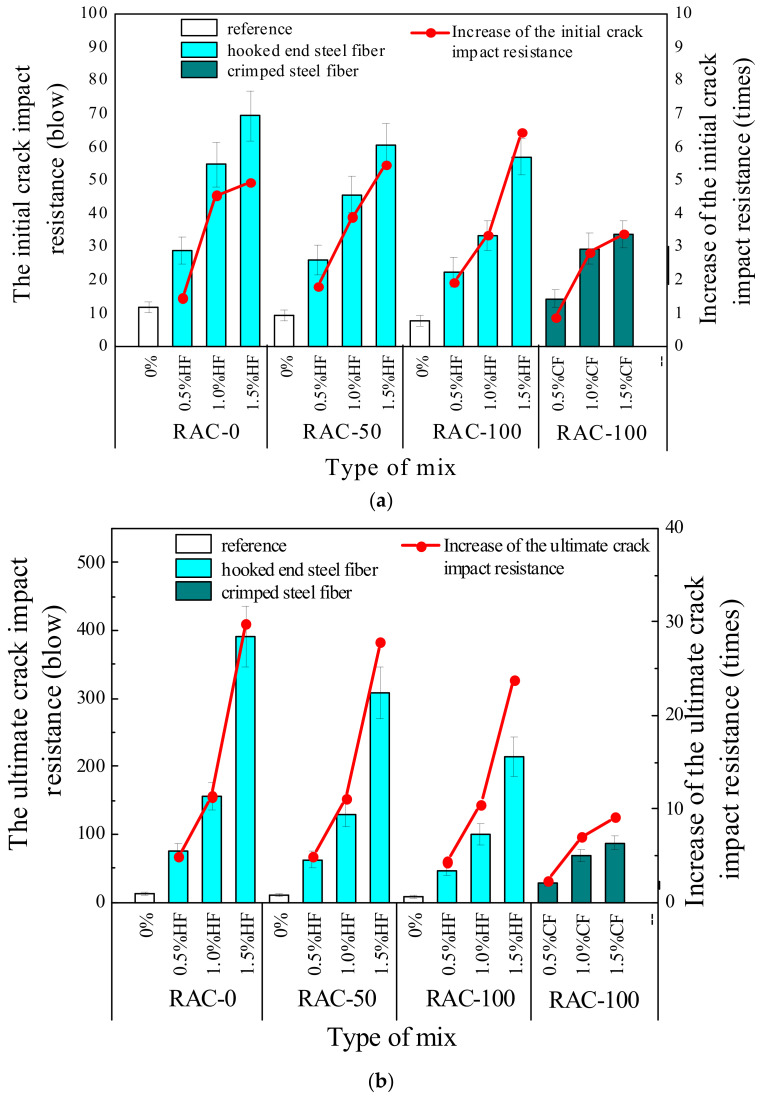
Test results of impact resistance: (**a**) the initial crack impact resistance; (**b**) the ultimate crack impact resistance; (**c**) The percentage increase in the number of post-initial-crack blows until failure.

**Figure 10 materials-15-07029-f010:**
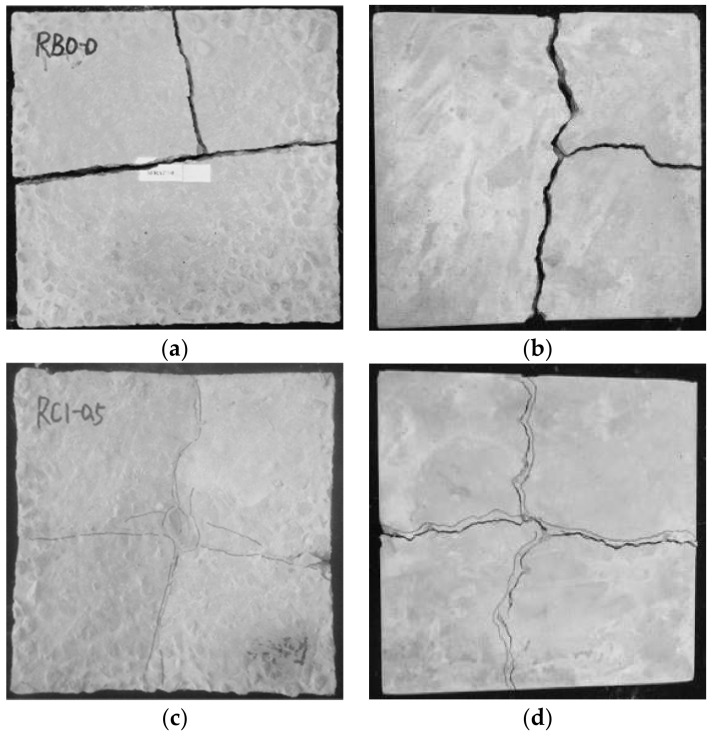
Comparison of impact failure modes of specimens: (**a**) R50-H0 (front); (**b**) R50-H0 (back); (**c**) R100-H0.5 (front); (**d**) R100-H0.5 (back); (**e**) R100-C0.5 (front); (**f**) R100-C0.5 (back).

**Figure 11 materials-15-07029-f011:**
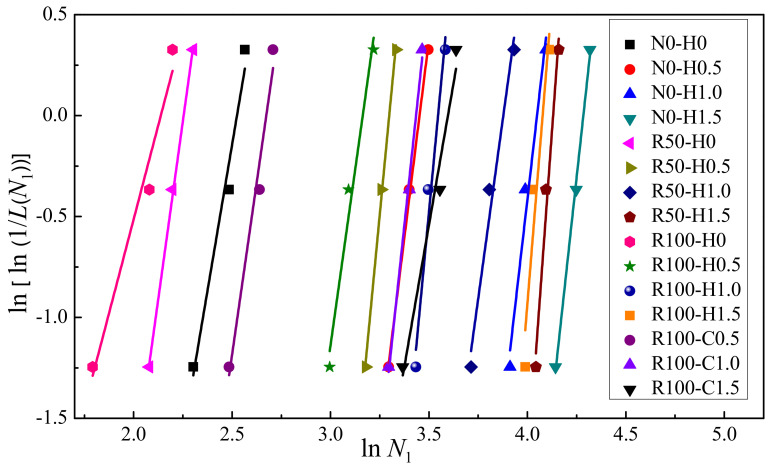
Linear regression of *N*_1_ in Weibull distribution.

**Figure 12 materials-15-07029-f012:**
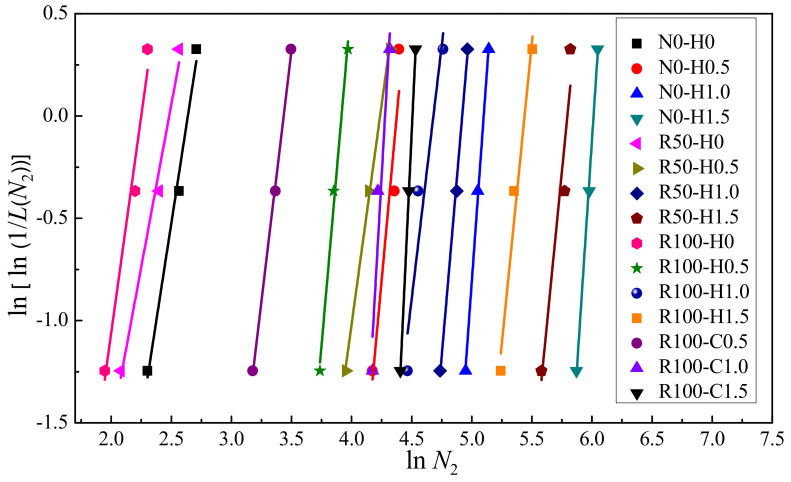
Linear regression of *N*_2_ in Weibull distribution.

**Figure 13 materials-15-07029-f013:**
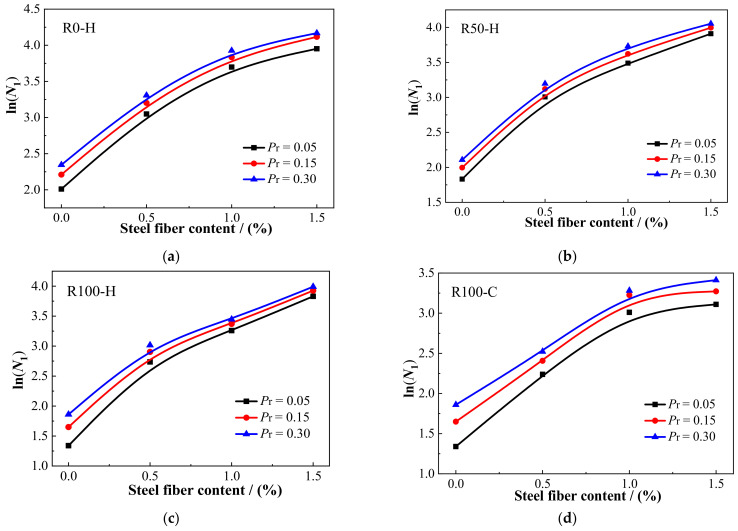
The curves of *P*_r_–*N*_1_–*V*_f_: (**a**) R0-H; (**b**) R50-H; (**c**) R100-H; (**d**) R100-C.

**Figure 14 materials-15-07029-f014:**
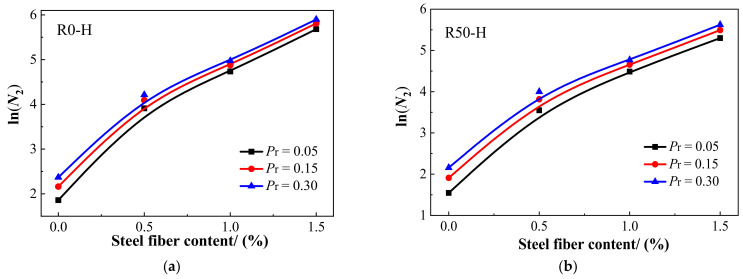
The curves of *P*_r_–*N*_2_–*V*_f_: (**a**) R0-H; (**b**) R50-H; (**c**) R100-H; (**d**) R100-C.

**Figure 15 materials-15-07029-f015:**
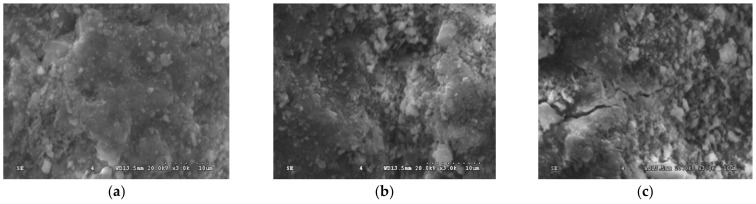
Microstructure of cement paste: (**a**) R0-H0; (**b**) R50-H0; (**c**) R100-H0.

**Figure 16 materials-15-07029-f016:**
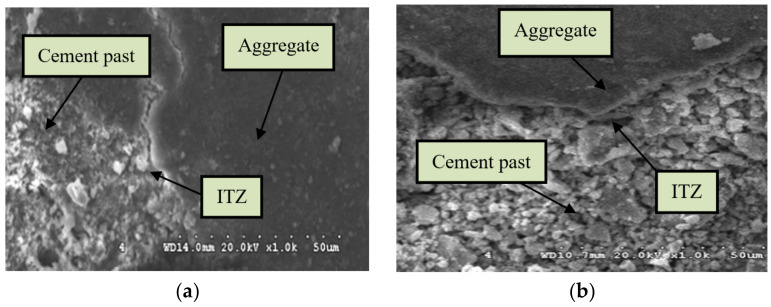
Microstructure of ITZ between aggregate and cement paste: (**a**) R0-H0; (**b**) R100-H0.

**Figure 17 materials-15-07029-f017:**
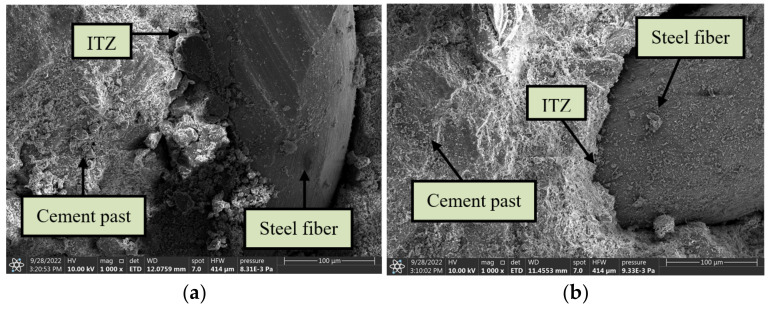
Microstructure of ITZ between fiber and cement paste: (**a**) R100-H1.0; (**b**) R100-C1.0.

**Table 1 materials-15-07029-t001:** General properties of ordinary Portland cement.

Chemical Characteristics	Result	Physical Characteristics	Result
SiO_2_	22.5%	Specific gravity	3.11
Al_2_O_3_	5.02%	Specific surface	358 m^2^/kg
Fe_2_O_3_	4.28%	Soundness	Qualified
CaO	61.85%	Initial setting time	175 min
MgO	1.58%	Final setting time	316 min
SO_3_	2.35%	3 day compressive strength	24.2 MPa
Na_2_O	0.23%	28 day compressive strength	49.5 MPa
K_2_O	1.05%	3 day flexural strength	5.1 MPa
Loss on Ignition	1.14%	28 day flexural strength	8.5 MPa

**Table 2 materials-15-07029-t002:** Properties of aggregates.

Property	Fine Aggregate	NCA	RCA
Specific gravity	2.67	2.74	2.28
Bulk density (kg/m^3^)	1555	1461	1226
Saturated surface dry water absorption (%)	1.9	1.7	6.8
Crush index (%)	-	8.1	17.1
Maximum aggregate size (mm)	4.75	20	20
Minimum aggregate size (mm)	-	5	5

**Table 3 materials-15-07029-t003:** Properties of steel fibers.

Fiber Types	Length (mm)	Diameter (mm)	Aspect Ratio	Density (kg/m^3^)	Tensile Strength (MPa)
Hooked end	30	0.75	40	7850	1100
Crimped	30	1.29	23	7850	990

**Table 4 materials-15-07029-t004:** Mix proportions of the concrete mixture.

Mixture	Cement (kg/m^3^)	Sand (kg/m^3^)	Water (kg/m^3^)	RCA (kg/m^3^)	NCA (kg/m^3^)	HF *V*_f_ (%)	CF *V*_f_ (%)	Superplasticizer (kg/m^3^)
R0-H0	410	719	166	0	1073	0	0	2
R0-H0.5	410	719	166	0	1073	0.5	0	2
R0-H1.0	410	719	166	0	1073	1.0	0	2
R0-H1.5	410	719	166	0	1073	1.5	0	2
R50-H0	410	719	166	536	536	0	0	2
R50-H0.5	410	719	166	536	536	0.5	0	2
R50-H1.0	410	719	166	536	536	1.0	0	2
R50-H1.5	410	719	166	536	536	1.5	0	2
R100-H0	410	719	166	1073	0	0	0	2
R100-H0.5	410	719	166	1073	0	0.5	0	2
R100-H1.0	410	719	166	1073	0	1.0	0	2
R100-H1.5	410	719	166	1073	0	1.5	0	2
R100-C0.5	410	719	166	1073	0	0	0.5	2
R100-C1.0	410	719	166	1073	0	0	1.0	2
R100-C1.5	410	719	166	1073	0	0	1.5	2

**Table 5 materials-15-07029-t005:** Test items, dimensions, and numbers of specimens.

Test Items	Dimensions (mm)	Number of Specimens
Compressive strength	150 × 150 × 150	45
Splitting tensile strength	150 × 150 × 150	45
Flexural performance	100 × 100 × 400	45
Impact strength	300 × 300 × 50	45

**Table 6 materials-15-07029-t006:** The impact resistance of initial crack for steel fiber-reinforced RAC.

Mix	*N* _1_						Initial Crack Energy (J)
	No. 1	No. 2	No. 3	Mean	SD	COV (%)	
R0-H0	10	12	13	12	1.53	13	105.8
R0-H0.5	25	28	33	29	4.04	14	255.8
R0-H1.0	49	53	62	55	6.66	12	485.1
R0-H1.5	62	69	77	69	7.51	11	608.6
R50-H0	8	9	11	9	1.53	16	79.4
R50-H0.5	22	25	31	26	4.58	18	229.3
R50-H1.0	41	44	52	46	5.69	12	405.7
R50-H1.5	53	62	66	60	6.66	11	529.2
R100-H0	6	8	9	8	1.53	20	70.6
R100-H0.5	18	22	27	22	4.51	20	194.0
R100-H1.0	29	33	38	33	4.51	14	291.1
R100-H1.5	52	56	63	57	5.57	10	502.7
R100-C0.5	12	14	17	14	2.52	18	123.5
R100-C1.0	24	31	33	29	4.73	16	255.8
R100-C1.5	29	35	37	34	4.16	12	299.9

SD—standard deviation; COV—coefficient of variation.

**Table 7 materials-15-07029-t007:** The impact resistance of final crack for steel fiber-reinforced RAC.

Mix	*N* _2_						Ultimate Failure Energy (J)
	No. 1	No. 2	No. 3	Mean	SD	COV (%)	
R0-H0	11	12	15	13	2.08	16	114.7
R0-H0.5	84	62	78	75	11.37	15	661.5
R0-H1.0	136	156	177	156	20.50	13	1375.9
R0-H1.5	342	401	429	391	44.41	11	3448.6
R50-H0	8	11	13	11	2.52	24	97.0
R50-H0.5	52	75	63	63	11.50	18	555.7
R50-H1.0	147	112	129	129	17.50	14	1137.8
R50-H1.5	65	321	337	308	37.81	12	2716.6
R100-H0	7	9	10	9	1.53	18	79.4
R100-H0.5	49	38	53	47	7.77	17	414.5
R100-H1.0	87	95	117	100	15.53	16	882.0
R100-H1.5	189	211	246	215	28.75	13	1896.3
R100-C0.5	24	29	33	29	4.51	16	255.8
R100-C1.0	79	68	61	69	9.07	13	608.6
R100-C1.5	88	77	98	88	10.50	12	776.2

SD—standard deviation; COV—coefficient of variation.

**Table 8 materials-15-07029-t008:** Linear regression of resistance in Weibull distribution.

Impact Resistance Factor	Concrete Type	Regression Coefficient *α*	Regression Coefficient *β*	*R* ^2^
*N* _1_	R0-H0	5.793	14.625	0.953
	R0-H0.5	5.561	19.050	0.940
	R0-H1.0	6.356	25.840	0.871
	R0-H1.5	7.254	31.150	0.989
	R50-H0	4.779	11.060	0.907
	R50-H0.5	4.443	14.860	0.911
	R50-H1.0	6.172	23.980	0.827
	R50-H1.5	6.867	28.550	0.939
	R100-H0	3.724	7.960	0.943
	R100-H0.5	3.877	12.420	0.989
	R100-H1.0	5.804	20.750	0.983
	R100-H1.5	7.983	32.680	0.922
	R100-C0.5	4.475	12.310	0.964
	R100-C1.0	4.503	15.600	0.859
	R100-C1.5	6.002	21.500	0.891
*N* _2_	R0-H0	4.682	12.270	0.806
	R0-H0.5	4.858	21.340	0.906
	R0-H1.0	5.974	30.540	0.996
	R0-H1.5	6.685	40.292	0.949
	R50-H0	3.176	7.880	0.976
	R50-H0.5	4.299	18.210	0.997
	R50-H1.0	5.789	28.540	0.996
	R50-H1.5	5.989	34.710	0.865
	R100-H0	4.243	9.550	0.947
	R100-H0.5	4.414	17.350	0.949
	R100-H1.0	4.951	23.170	0.838
	R100-H1.5	5.874	31.950	0.947
	R100-C0.5	4.917	16.890	0.997
	R100-C1.0	5.992	25.780	0.949
	R100-C1.5	6.523	29.580	0.999

**Table 9 materials-15-07029-t009:** Impact resistance times of SFRAC under different probabilities of failure.

Mix No.	Probabilities of Failure (*P*_r_)											
	5%		10%		15%		20%		25%		30%	
	*N* _1_	*N* _2_	*N* _1_	*N* _2_	*N* _1_	*N* _2_	*N* _1_	*N* _2_	*N* _1_	*N* _2_	*N* _1_	*N* _2_
R0-H0	7	8	8	8	9	10	9	10	10	10	10	11
R0-H0.5	18	44	21	51	22	56	23	59	25	63	26	65
R0-H1.0	37	101	41	114	44	122	46	129	48	135	50	140
R0-H1.5	49	266	54	296	57	316	60	331	62	344	64	355
R50-H0	5	5	6	6	7	7	7	7	8	8	8	9
R50-H0.5	15	35	17	41	19	45	20	49	21	52	22	54
R50-H1.0	30	83	34	94	36	101	38	107	40	112	41	116
R50-H1.5	41	200	46	226	49	243	51	256	53	267	55	277
R100-H0	4	5	5	6	5	6	6	7	6	7	6	8
R100-H0.5	11	26	14	31	15	34	17	36	18	38	19	40
R100-H1.0	21	59	24	68	26	75	28	80	29	84	30	87
R100-H1.5	41	139	45	157	48	169	50	178	51	186	53	193
R100-C0.5	8	17	9	20	10	21	11	23	12	24	12	25
R100-C1.0	17	45	19	51	21	55	23	58	24	60	25	62
R100-C1.5	22	59	25	66	27	71	28	74	29	77	30	80

## Data Availability

All relevant data generated by the authors or analyzed during the study are included within the paper.

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
