# Peer review of "The Impact Resistance and Mechanical Properties of Recycled Aggregate Concrete with Hooked-End and Crimped Steel Fiber"

_materials, 2022, doi:10.3390/ma15197029_

Round 1
Reviewer 1 Report
The author(s) presented the experimental and statistical results on the impact and mechanical properties of steel fibre reinforced recycled concrete aggregate concretes. However, there are a few fundamental mistakes in the microstructural analysis.
Detailed comments are given below:
1. To date, there is one article (Characterization and Optimization of Mechanical and Impact Properties of Steel Fiber Reinforced Recycled Concrete, from Omidinasab et al., 2022, in International Journal of Civil Engineering) published on similar scopes as this paper. Highly recommending the author(s) to update the research background and discussions to distinguish this study with the published article. Also, this article made the statement (line 105-106) “to the best of the au-104 thor's knowledge, the investigation that compares different steel fiber shape (e.g., HF, CF) on mechanical properties and impact resistance of SFRAC has not been reported and needs special emphasis” to be inappropriate and needs revision. Also, I would like to suggest to further elaborate the research gaps of what does it means on “needs special emphasis” in this statement.
2. Table 2: Since the main scope of the paper lies on the impact resistance of the RCA concrete, thus I am suggesting if the author(s) can conduct the aggregate stiffness/impact resistance of the NCA and RCA such as aggregate crushing value and aggregate impact value, as both values are directly related to compressive strength and impact resistance of RCA concrete, respectively, and subsequently supplement the discussions in line 241-243 (This phenomenon may be caused by the lower apparent density, higher crushing index (Zong et al., 2021) and water absorption (Ali and Qureshi, 2019) of RCA weaken the bearing capacity of RAC matrix.)
3. Figure 8, 10 and 11: Suggest to remove the regression equation in Figure 8, 10 and 11, as the number of data points is considered low to be adequate for an equation proposal. Also, the equation is invalid for other mix proportions (and only limit to the mix design in this paper) as there is no validation with other studies.
4. Figure 19: please explain on how can the author(s) identify the ITZ from the SEM images? Also, please explain (and/or aided labelling in Figure 19) on how to obtain the observation “ITZ between cement paste and RCA was more obvious than that of NCA specimen”, as the ITZ is not clear to be distinguished from Figure 19.
Also, the WD from both SEM images from Figure 19 are different, the magnification is the same, but at different WD (working distance) or focal distance, there will be difference in the feature observed in which the scale bar also different. Please provide the SEM images with the same WD in order for precise comparison of ITZ.
5. Figure 20: How can the author(s) can identify the steel fibre as labelled?
6. There is no linkages on the microstructure to the impact/mechanical properties of the concrete. Suggest the author(s) to use the microstructural analysis to supplement the impact/mechanical results.
Author Response
请参阅附件。

Reviewer 2 Report
This study investigates recycled aggregate concrete's impact resistance and mechanical properties with two different steel fibres (hooked end and crimped). This study considered two different fibres and two different percentages of natural aggregate replacement. Besides, a two-parameter Weibull distribution is used to analyze the variations in test results. This article requires some improvements before it can be accepted for publication,
1. Abstract: key findings presented here is general. I suggest quantifying the results and adding key findings.
2. Most of the literatures focused on the mechanical properties of concrete comprising steel fibres and RAC. However, the literature regarding the impact resistance performance of concrete steel fibre and RAC is lacking. It is recommended to add a few literature related to this research's objective.
3. Line 128-129, “polycarboxylate type superplasticizer, whose water reducing ratio was 20%~30%”. How were these ratios determined?. These ratios are in contrast with the literature?. Generally, the water reducing admixture is 0.5-4% of the cement weight. In this study, the water reducing admixture is 20-30%. Justify this by comparing it with Table 4.
4. Table 3, please check the tensile strength of crimped steel fibre 590 MPa, almost equal to the polypropylene fibre. The change in tensile strength leads to a huge difference in impact strength.
5. According to the ACI 544.2R-89, 1996, a steel ball of 4.54kg dropped from a height of 457 mm. However, in this study, a 3 kg steel ball repeatedly free fall from 300 mm. Justify why the drop weight and height deviated from the ACI 544.2R-89 guidelines.
6. Line 218: check the unit for the gravity acceleration (9.8 N/kg).
7. Figures 5, 7 and 9, the tensile strength of fibre plays an important role in increasing the compressive strength, splitting tensile strength and flexural strength of concrete. Since the tensile strength of HF is almost doubled, the CF. How do HF and CF-based concrete compressive, splitting tensile strength and flexural strength show a marginal difference?. Please justify this.
8. Figures 8,10 and 11, what is the rationale of linear equations?. How many data points are required to propose a relationship according to statistical analysis?
9. When you use statistical analysis (Weibull distribution), it is necessary to show the results of all specimens. How many number specimens per mixture tested?. I suggest you add all the specimen's values in Table 6.
10. The coefficient of variance in Table 6 ranged from 11 to 20% for N1 and 12 to 24% for N2. The determined coefficient is less, and what is the purpose of applying Weibull distribution to the impact test data?. Justify this.
11. It is recommended to mention the reasons for the variations in impact test results.
12. Compare the experimental average values of impact test results and statistical analysis results with different reliability levels to prove the effectiveness of the Weibull distribution.
Reviewer 3 Report
Interesting work and a well-prepared paper.
My only concern is acronym abuse - if someone reads the conclusions first - would they understand them without having to hunt the paper for definitions of acronyms
Round 2
Reviewer 2 Report
All comments are addressed sufficiently.